# Use of an Insulation-Tipped Knife during Rigid Bronchoscopic Treatment of Benign Tracheobronchial Stenosis

**DOI:** 10.3390/medicina57030251

**Published:** 2021-03-08

**Authors:** Ji-Won Park, Yousang Ko, Changhwan Kim

**Affiliations:** 1Department of Internal Medicine, Hallym University Sacred Heart Hospital, Anyang 14068, Korea; miunorijw@hallym.or.kr; 2Department of Pulmonary and Critical Care Medicine, Kangdong Sacred Heart Hospital, Seoul 05355, Korea; koyus@naver.com; 3Department of Internal Medicine, Jeju National University Hospital, Jeju National University School of Medicine, Jeju 63241, Korea

**Keywords:** benign tracheobronchial stenosis, complications, IT knife, rigid bronchoscopy

## Abstract

*Background and Objectives:* Tracheal or bronchial tears are potential complications of rigid bronchoscopy. This study aimed to investigate the acute complications and outcomes of using an insulation-tipped (IT) knife in combination with rigid bronchoscopic dilatation for treating benign tracheobronchial stenosis. *Materials and Methods:* We conducted a chart review of patients with benign tracheobronchial stenosis who were treated with rigid bronchoscopy and an IT knife at two referral centers. Treatment success was defined as a clinically stable state without worsening symptoms after 3 months of treatment. *Results:* Of the 23 patients with benign tracheobronchial stenosis, 15 had tracheal stenosis and 6 had main bronchial stenosis. Among them, three cases were of simple stenosis (13%), while the others were of complex stenosis (87%). The overall treatment success rate was 87.0%. Pneumomediastinum and subcutaneous emphysema occurred due to bronchial laceration in two cases of distal left main bronchial stenosis (8.7%), and no other significant acute complications developed. Silicone stents were inserted in 20 patients, and successful stent removal was possible in 11 patients (55.0%). Six of the seven stents inserted in patients with post-intubation tracheal stenosis were removed successfully (85.7%). However, most of the patients with post-tracheostomy tracheal stenosis required persistent stenting (80%). Pulmonary function was significantly increased after treatment, and the mean increase in the forced expiratory volume in 1 s was 391 ± 171 mL (160–700 mL). *Conclusion:* The use of an IT knife can be suggested as an effective and safe modality for rigid bronchoscopic treatment of benign tracheobronchial stenosis.

## 1. Introduction

Benign tracheobronchial stenosis is a challenging medical condition that may incur progressive dyspnea and life-threatening hypoxemia. The common causes of benign tracheobronchial stenosis are endotracheal intubation, tracheostomy, and post-operative [1]. Post-tuberculosis tracheobronchial stenosis is also a common cause in tuberculosis endemic areas [2].

The initial treatment of choice for benign tracheal stenosis was segmental tracheal resection with end-to-end anastomosis [3,4]. However, after Dumon developed a molded silicone stent with multiple external studs in 1987, stent insertion became a more reasonable therapeutic option [5]. In cases of post-intubation tracheal stenosis (PITS) and post-tracheostomy tracheal stenosis (PTTS), bronchoscopic intervention is preferred for simple stenosis, and tracheal sleeve resection may be needed for more complex lesions [6,7]. There are no clear criteria for selecting treatment modalities of benign bronchial stenosis. Bronchoscopic dilatation of bronchial stenosis is more challenging than that of tracheal stenosis because of the smaller luminal diameter and acute angulation of stenotic bronchus. Therefore, surgical bronchoplasty may be considered if anatomically feasible [8].

Before performing bronchoscopic dilatation of benign tracheobronchial stenosis using a rigid bronchoscope and/or balloon, laser or electrocautery can be used to cut the stenotic lesions with dense fibrosis to minimize injury of adjacent normal tracheobronchial mucosa. However, tracheal or bronchial tears are potential complications of rigid bronchoscopy, which may result in pneumothorax, pneumomediastinum, or malpositioning of the stent [9]. Electrocautery has advantages over laser in terms of lower cost and absence of associated biohazard; however, its efficacy is similar to that of laser [10]. Although the use of an electrocautery knife is usually safe, it should be applied carefully in the stenotic trachea or bronchus because the site distal to the narrowed lesion is seldom visible and is exposed to the risk of injury to normal airway structures.

The insulation-tipped (IT) diathermic knife is an endoscopic instrument with a small ceramic ball attached as an insulator to the end of a needle knife to prevent perforation (Figure 1). This study aimed to investigate the acute complications and outcomes of using an IT knife in combination with rigid bronchoscopic dilatation for treating benign tracheobronchial stenosis.

## 2. Materials and Methods

### 2.1. Patients

We retrospectively conducted a chart review of patients with benign tracheobronchial stenosis who were treated with rigid bronchoscopy and an IT knife at two referral centers in Korea between July 2013 and May 2019.

The protocol for this study was approved by the Institutional Review Board of each participating hospital (Jeju National University Hospital and Kangdong Sacred Heart Hospital) for the review and publication of information from the patient records. Requirement for informed consent was waived because of the retrospective nature of this study.

### 2.2. Airway Intervention Techniques

Airway anatomy was evaluated using chest computed tomography and, when possible, flexible bronchoscopy. Airway intervention was performed according to the standard techniques described by Colt and Dumon and Kim [5,11]. In short, patients were intubated with a rigid bronchoscope (Karl-Storz, Tuttlingen, Germany), and a flexible bronchoscope (BF 1T260, Olympus, Tokyo, Japan) was introduced through rigid bronchoscope after induction of general anesthesia.

An IT knife (Olympus, Tokyo, Japan) was inserted through a 2.8 mm working channel of the BF 1T260 and manipulated to cut the stenotic lesions with dense fibrosis using the current delivered from the high frequency electrosurgical system (VIO 300 D, ERBE, Tübingen, Germany) (Figure 2). Subsequently, various combinations of airway intervention techniques were used, including dilatation with a balloon (CRE balloon, Boston Scientific Corporation, Marlborough, MA, USA), mechanical bougienage, and the insertion of silicone stents, depending on the characteristics of the tracheobronchial stenosis.

### 2.3. Airway Stents

The types of airway silicone stents used during the study included the Natural stent (M1S Co., Seoul, Korea) and Dumon stent (Novatech, La Ciotat, France). The Natural stent is a silicone stent developed at Samsung Medical Center in 2002. A study in patients with benign tracheobronchial stenosis showed that the Natural stent was as effective and safe as the Dumon stent [12]. However, the production of Natural stents ceased due to commercial issues. Dumon stents have been commercially available for medical use in South Korea since 2015.

### 2.4. Data Collection

We collected baseline data on patient demographics, comorbidities, etiology of tracheobronchial stenosis, and site and nature of stenosis. Data on applied airway intervention techniques, acute complications, pre- and post-treatment pulmonary functions, and treatment outcomes were also collected. The nature of tracheobronchial stenosis was categorized as simple or complex according to a previous report. Simple stenosis was defined as lesions with an endoluminal occlusion of a short segment (<1 cm), with the absence of malacia or loss of cartilaginous support. Complex stenosis was defined as stenosis with extensive scarring (≥1 cm) and varying degrees of cartilage involvement or circumferential contraction scarring or as stenosis associated with malacia and inflammation [7]. Treatment outcomes included acute complications, treatment success, successful stent removal, persistent stent placement, change of pulmonary function, need for surgical management, and mortality. Treatment success was defined as a clinically stable state without worsening symptoms after 3 months of treatment.

### 2.5. Statistical Analysis

Continuous variables are presented as mean ± standard deviation (SD), and categorical variables are presented as numbers (percentages). Spirometry data before and after the procedure were compared using a Wilcoxon signed-rank test. A value of *p* < 0.05 was considered statistically significant. All statistical analyses were performed using the statistical software package IBM SPSS Statistics v23.0.0.0 (IBM Co., Armonk, NY, USA).

## 3. Results

### 3.1. Baseline Characteristics

During the study period, 23 patients with benign tracheobronchial stenosis were treated with rigid bronchoscopy and an IT knife. Their mean age was 54.2 ± 14.8 years (range, 19–79 years), and nine patients (39.1%) were males. The site and etiology of stenosis included fifteen tracheal stenosis (nine PITS and six PTTS), six main bronchial stenosis (five post-infectious and one post-operative), and two stenosis in the right bronchus intermedius (two post-infectious). In terms of the nature of stenosis, only three cases were classified as simple stenosis (13%), while the others were complex stenosis (87%). The mean length of stenosis was 32.6 ± 12.4 mm (range, 10–50mm). One patient with PITS and four patients with post-infectious stenosis had no comorbidities; however, the others had 1–5 concomitant diseases (Table 1).

### 3.2. Treatment Modalities

The mean follow-up duration was 33.7 ± 24.5 months (range, 2–83 months). Twenty patients (87.0%) required mechanical bougienage and silicone stent insertion, and 7 of 20 patients needed additional interventional bronchoscopies due to stent migration and/or granulation tissue formation around the stents. Balloon dilatation was performed adjunctively in three patients with bronchial stenosis for distal airway narrowing. One patient required an uncuffed tracheostomy after the intervention because of his bedridden status.

### 3.3. Treatment Outcomes

The clinical course and treatment outcomes are shown in Figure 3 and Table 2. The overall treatment success rate was 87.0%. Three patients (two PITS and one PTTS) were successfully treated using an IT knife without stent insertion. Of the twenty patients with silicone stent insertion, unsuccessful stent removal occurred in three patients within 2 months: two patients had bronchial stenosis (one left main and one right bronchus intermedius stenosis), and one patient had complex mixed PTTS who eventually underwent tracheal resection and end-to-end anastomosis. Successful stent removal was possible in eleven patients (55%), and six patients required persistent stent insertion (30%). In patients with PITS, six of the seven stents inserted were removed successfully (85.7%). However, four of five PTTS patients with stent insertion needed persistent stenting (80.0%), and one patients with PTTS patient required surgical treatment. During the follow-up period, two patients with a stent in situ died of variceal bleeding and myocardial infarction (one PITS and one PTTS).

Pneumomediastinum and subcutaneous emphysema occurred as an acute complication in two cases of distal left main bronchial stenosis (8.7%). Of these, one patient had long left main bronchial stenosis with an acute angular direction from the trachea, which narrowed into the left second carina. Therefore, bronchial laceration occurred unavoidably during the insertion of the rigid bronchoscope. A silicone stent was inserted after the laceration was healed, but could not be maintained owing to the stenosis distal to the stent. The other patient also had a cicatrical stricture in the left main bronchus, and a small bronchial tear developed during mechanical bougienage. However, a silicone stent was inserted at the same time and successfully removed in 20 months. No other significant acute complications occurred.

Pulmonary function test results were available for 12 patients without tracheostomy before treatment and 11 patients after treatment. Among the 11 patients who had both test results, there were statistically significant increases in the forced expiratory volume in 1 s (FEV_1_) (from mean 1.65 ± 0.40 to 2.04 ± 0.42 L, Z = −2.934, *p* = 0.003) and forced vital capacity (from mean 2.47 ± 0.72 to 2.85 ± 0.63 L, Z = −2.937, *p* = 0.003). The mean increase in FEV_1_ was 391 ± 171 mL after treatment (range, 160–700 mL).

## 4. Discussion

The present study aimed to investigate the acute complications and treatment outcomes of using an IT knife with rigid bronchoscopy for treating benign tracheobronchial stenosis. Most of the patients had complex stenosis and were treated successfully. Bronchial laceration was an acute complication in only two cases with left main bronchial stenosis caused by a previous infection. In cases of tracheal stenosis, most of the inserted stents were successfully removed in PITS but needed to be retained in PTTS. Pulmonary function significantly improved after treatment.

Thus far, there have been only two reports on the use of an IT knife in bronchoscopic treatment. One is a case report on using an IT knife for the treatment of PITS [13]. The other is a comparison of effectiveness between an electrosurgical knife and laser for the treatment of benign tracheobronchial stenosis; however, the needle-type knife was mainly used rather than an IT knife [14]. Moreover, all previously reported cases that were treated with an IT knife had simple web-like benign stenosis, and the suggestions were focused on the benefits of an electrosurgical knife over laser in short segment stenosis. However, most patients in the present study had complex stenosis, and our primary concern was to lessen the likelihood of injury to the normal tracheobronchial structures surrounding the stenotic lesion by using an IT knife during rigid bronchoscopic treatment.

The IT knife was developed by Ono et al. at the National Cancer Center Hospital in the late 1990s and has been used during endoscopic submucosal dissection [15]. An insulated small ceramic ball attached to the tip of a high frequency needle knife allows safe and easy incision of the tissue and reduces the risk of perforation (Figure 1). Although the complications of electrocautery generally include bleeding and airway perforation [16], an IT knife has a lateral cutting capability with a lower risk of distal airway injury due to the insulating tip. In addition, electrocautery is widely available in most medical facilities, and the combined use of an IT knife with electrocautery is more cost-effective than laser treatment is. A previous study reported that the use of an electrosurgical knife with the intermittent cutting mode caused fewer fibrin plates in comparison with that caused by the use of a laser [14]. Therefore, we believe that an IT knife is an optimal cutting tool for short-sighted tracheobronchial stenosis with a small diameter, which are difficult to visualize.

Ballooning and/or mechanical bougienage are commonly performed to dilate complex tracheobronchial stenosis. Along with mechanical bougienage, the main side effect of balloon bronchoplasty is partial or complete airway laceration resulting in pneumothorax, pneumomediastinum, and hemorrhage [16,17]. If an airway injury occurs, granulation tissue and stricture develop within the first four weeks and usually lead to symptoms, signs, and radiological changes [18]. Therefore, in most cases of complex stenosis, silicone stents are inserted to maintain airway patency against granulation tissue formation, fibrotic stricture, and malacia after dilatation. Mechanical bougienage by rigid bronchoscope is a prerequisite for silicone stent deployment. We believe that cutting the dense stenosis using an IT knife can facilitate easy advancement of the rigid bronchoscope through the stenotic lesion with a lesser risk of airway injury. In a retrospective analysis of rigid bronchoscopic intervention using the standard techniques for 80 cases of post-tuberculosis tracheobronchial stenosis, eight patients (10%) experienced acute complications such as massive bleeding, pneumomediastinum, and pneumothorax requiring chest tube placement [2]. Although it cannot be compared directly, only two cases of pneumomediastinum occurred as an acute complication (8.7%) in the present study, which were easily resolved by oxygen supplement. In addition, ballooning before mechanical bougienage was seldom needed because the loosening of dense stenosis could be secured to perform bougienage by cutting with an IT knife.

The present study has several limitations. The study was conducted in a retrospective manner, and we could not perform a comparison with another treatment. Moreover, the study subjects were heterogeneous in terms of the site, reason, and nature of stenosis. Acute complications developed only in cases of left main bronchial stenosis, mainly due to the challenging bronchial structure. To adequately evaluate the effectiveness of an IT knife in reducing acute complications and the adjunctive use of a balloon during the rigid bronchoscopic treatment, further studies with a larger number of homogenous patients are warranted.

## 5. Conclusions

The present study suggests that the combined use of an IT knife with rigid bronchoscopy may be an effective and relatively safe treatment modality for benign complex tracheobronchial stenosis. This technique may help in loosening the dense fibrotic stenosis and facilitate mechanical bougienage with a lower risk of airway injury.

## Figures and Tables

**Figure 1 medicina-57-00251-f001:**
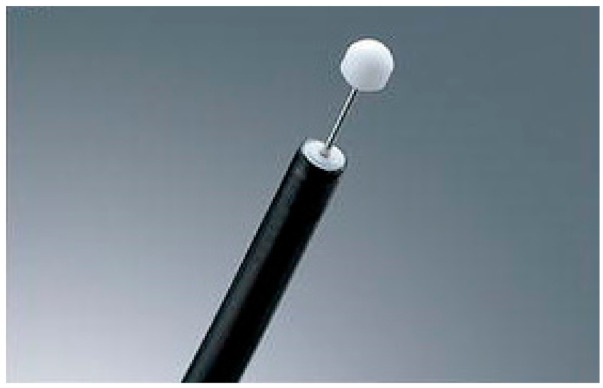
The insulation-tipped (IT) diathermic knife (KD-610L, Olympus, Tokyo, Japan) has a small ceramic ball attached as an insulator to the end of a needle knife to prevent perforation.

**Figure 2 medicina-57-00251-f002:**
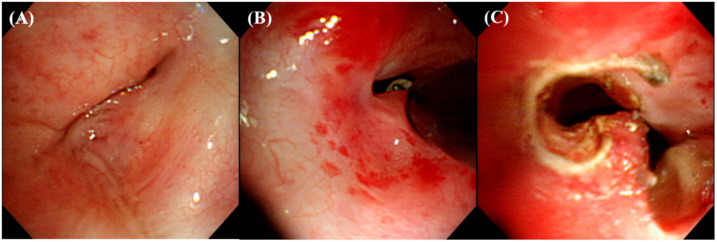
(**A**) A brochoscopic image of a case of fibrostenosis at the proximal left main bronchus. (**B**,**C**) Example images of cutting the dense fibrotic lesion with an IT knife and electrosurgical system.

**Figure 3 medicina-57-00251-f003:**
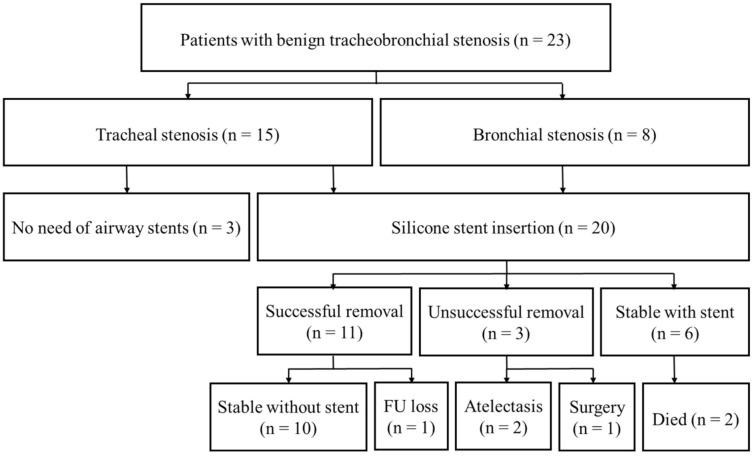
Clinical course of patients with benign tracheobronchial stenosis treated with an IT knife and rigid bronchoscopy (FU: follow-up).

**Table 1 medicina-57-00251-t001:** Characteristics of the study patients.

	Study Patients (*n* = 23)	
Age (years)	54.2 ± 14.8	
Sex (male/female)	9/14 (39.1%/60.9%)	
Comorbidities	Respiratory diseases	8 (34.8%)
	Chronic renal failure	5 (21.7%)
	Diabetes mellitus	5 (21.7%)
	Ischemic heart disease	4 (17.4%)
	Hypertension	4 (17.4%)
	Heart failure	3 (13.0%)
	Chronic liver diseases	3 (13.0%)
	Others *	8
No. of comorbidities	1.8 ± 1.44	
Site of stenosis	Trachea	15 (65.2%)
	Main bronchus	6 (26.1%)
	Right bronchus intermedius	2 (8.7%)
Reason of stenosis	PITS	9 (39.1%)
	PTTS	6 (26.1%)
	Post-infectious	7 (30.4%)
	Post-operative	1 (4.3%)
Nature of stenosis ^†^	Simple	3 (13.0%)
	Complex	20 (87.0%)
Length of stenosis (mm)	32.6 ± 12.4	
Stent insertion	20 (87.0%)	

Data are presented as *n* (%) or mean ± standard deviation. PITS = post-intubation tracheal stenosis; PTTS = post-tracheostomy tracheal stenosis. * Others: arrhythmia (3), depressive disorder (2), hypothyroidism (1), tuberculous lympadenitis (1), Lennox-Gastaut syndrome (1). ^†^ Simple stenosis was defined as lesions with endoluminal occlusion of a short segment (<1 cm), with the absence of malacia or loss of cartilaginous support. Complex stenosis was defined as stenosis with extensive scarring (≥1 cm) and varying degrees of cartilage involvement or circumferential contraction scarring or stenosis associated with malacia and inflammation.

**Table 2 medicina-57-00251-t002:** Overall treatment outcomes.

	Patients (*n* = 23)
Acute complication	2 (8.7%)
Treatment success *	20 (87.0%)
Successful stent removal	11/20 (55%)
Stable with stent	6/20 (30%)
Surgical treatment	1 (4.3%)
Death	2 (8.7%)

Data are presented as *n* (%). * Treatment success was defined as a clinically stable state without worsening symptoms after 3 months of treatment.

## Data Availability

Data and material are available on reasonable request.

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
