# Peer review of "Use of an Insulation-Tipped Knife during Rigid Bronchoscopic Treatment of Benign Tracheobronchial Stenosis"

_medicina, 2021, doi:10.3390/medicina57030251_

Round 1
Reviewer 1 Report
Congratulations, I did not know about this knife.
I occupy myself with benign stenosis - and what you may add to your article is a discussion of plugging in permanent silicon stent need. This is why we fix these stents as we have cut them down to the stenosis length + 1cm on both ends. This has the advantage that you have less plugging problems.
I like this paper, really good job !
Author Response
Thank you for your interest and favorable comments. As you suggest, we usually place silicon stents with 1cm (or 0.5cm) longer on both ends than the stenotic lesions in case of tracheal (or bronchial) stenosis. In the present study, the rate of successful stent removal (55%) was similar with another study results of benign tracheobronchial stenosis (Respirology 2006; 11: 748), and this study aimed to investigate the acute complications and outcomes of using an IT knife in combination with rigid bronchoscopic dilatation. Therefore, we did not mention about the need of permanent stent in the discussion section.

Reviewer 2 Report
Overall this manuscript is well written and the study advances the field in terms of use of an insulation-tipped knife during rigid bronchoscopic treatment of benign tracheobronchial stenosis. The methodology is correct and the study and the data are well organized and commented. That said, I do have several concerns that I think if addressed may make this work stronger.
- In the conclusion of this study, the IT diathermic knife was a safe treatment modality in benign complex tracheobronchial stenosis while lowering the risk of airway injury. In the cited references, there are only case reports of successful treatment with an IT knife in benign simple stenosis. If there are any previous studies about the complication of treating complex tracheobronchial stenosis without the use of an IT knife, adding a comparison with those studies may help support the conclusion of this study.
- In cases of tracheal stenosis, most of the inserted stents were successfully removed in PITS but needed to be retained in PTTS. In addition to the reason for previous infection, I wonder if there were other reasons for differences in stent removal rate between PITS and PTTS, such as differences in sites of stenosis and treatment modalities.
- The percentage of ‘others’ in comorbidities section of Table 1 isn’t provided.
- Do you have any video clips of IT knife resection for airway stenosis? If you will provide it, it must be very helpful for international readers to understand how it works.
Author Response
Overall this manuscript is well written and the study advances the field in terms of use of an insulation-tipped knife during rigid bronchoscopic treatment of benign tracheobronchial stenosis. The methodology is correct and the study and the data are well organized and commented. That said, I do have several concerns that I think if addressed may make this work stronger.
Response:
Thank you for your interest and favorable comments. We have revised the manuscript according to your comments.
- In the conclusion of this study, the IT diathermic knife was a safe treatment modality in benign complex tracheobronchial stenosis while lowering the risk of airway injury. In the cited references, there are only case reports of successful treatment with an IT knife in benign simple stenosis. If there are any previous studies about the complication of treating complex tracheobronchial stenosis without the use of an IT knife, adding a comparison with those studies may help support the conclusion of this study.
Response:
We totally agree with your suggestion. We have added a comparison of the acute complications with a previous study into the discussion section as follows;
In a retrospective analysis of rigid bronchoscopic intervention using the standard techniques for 80 cases of post-tuberculosis tracheobronchial stenosis, eight patients (10%) experienced acute complications such as massive bleeding, pneumomediastinum, and pneumothorax requiring chest tube placement [2]. Although it cannot be compared directly, only two cases of pneumomediastinum occurred as an acute complication (8.7%) in the present study, which were easily resolved by oxygen supplement. In addition, ballooning before mechanical bougienage was seldom needed because the loosening of dense stenosis could be secured to perform bougienage by cutting with an IT knife.
- In cases of tracheal stenosis, most of the inserted stents were successfully removed in PITS but needed to be retained in PTTS. In addition to the reason for previous infection, I wonder if there were other reasons for differences in stent removal rate between PITS and PTTS, such as differences in sites of stenosis and treatment modalities.
Response:
We appreciate your question. There is an article that can be a suitable answer, entitled “Clinical significance of differentiating post-intubation and post-tracheostomy tracheal stenosis” (Respirology 2017; 22: 513–520). The authors revealed that there were significant differences between PITS and PTTS in terms of stenosis characteristics and clinical outcomes. Patients with PTTS had more complicated stenosis and a lower success rate for airway stent removal than those with PITS, which may reflect different mechanisms of tracheal injury caused by endotracheal tube or tracheostomy. Those findings are consistent with the results of our study, but we would not mention about the different rates of successful stent removal between PITS and PTTS in the discussion section because it is not something new and digresses from the main subject of our study.
- The percentage of ‘others’ in comorbidities section of Table 1 isn’t provided.
Response:
Thank you for pointing out the shortcomings. We have described other comorbidities in detail at the end of Table 1 as follows;
Table 1. Characteristics of the study patients.
|
|
Study patients (n = 23) |
|
|
Age (years) |
54.2 ± 14.8 |
|
|
Sex (male/female) |
9/14 (39.1%/60.9%) |
|
|
Comorbidities |
Respiratory diseases |
8 (34.8%) |
|
|
Chronic renal failure |
5 (21.7%) |
|
|
Diabetes mellitus |
5 (21.7%) |
|
|
Ischemic heart disease |
4 (17.4%) |
|
|
Hypertension |
4 (17.4%) |
|
|
Heart failure |
3 (13.0%) |
|
|
Chronic liver diseases |
3 (13.0%) |
|
|
Others* |
8 |
|
No. of comorbidities |
1.8. ± 1.44 |
|
|
Site of stenosis |
Trachea |
15 (65.2%) |
|
|
Main bronchus |
6 (26.1%) |
|
|
Right bronchus intermedius |
2 (8.7%) |
|
Reason of stenosis |
PITS |
9 (39.1%) |
|
|
PTTS |
6 (26.1%) |
|
|
Post-infectious |
7 (30.4%) |
|
|
Post-operative |
1 (4.3%) |
|
Nature of stenosis* |
Simple |
3 (13.0%) |
|
|
Complex |
20 (87.0%) |
|
Length of stenosis (mm) |
32.6 ± 12.4 |
|
|
Stent insertion |
20 (87.0%) |
|
Data are presented as n (%) or mean ± standard deviation. PITS = post-intubation tracheal stenosis; PTTS = post-tracheostomy tracheal stenosis *Simple stenosis was defined as lesions with endoluminal occlusion of a short segment (< 1 cm), with the absence of malacia or loss of cartilaginous support. Complex stenosis was defined as stenosis with extensive scarring (≥ 1 cm) and varying degrees of cartilage involvement or circumferential contraction scarring or stenosis associated with malacia and inflammation.
*Others: arrhythmia (3), depressive disorder (2), hypothyroidism (1), tuberculous lympadenitis (1), Lennox-Gastaut syndrome (1).
- Do you have any video clips of IT knife resection for airway stenosis? If you will provide it, it must be very helpful for international readers to understand how it works.
We agree with your suggestion. However, our centers have not been equipped with video recording devices as a part of the portable bronchoscopic system. I am very sorry for not showing any video clips. Instead, we have demonstrated representative images in figure 2.

Round 2
Reviewer 2 Report
All concerns were resolved clearly.
The results of the study will be very informative to international readers, especially interventional pulmonologist.